# Towards Unified Image Deblurring using a Mixture-of-Experts Decoder

## Abstract

Image deblurring, removing blurring artifacts from images, is a fundamental task in computational photography and low-level computer vision. Existing approaches focus on specialized solutions tailored to particular blur types, thus, these solutions lack generalization. This limitation in current methods implies requiring multiple models to cover several blur types, which is not practical in many real scenarios. In this paper, we introduce the first all-in-one deblurring method capable of efficiently restoring images affected by diverse blur degradations, including global motion, local motion, blur in low-light conditions, and defocus blur. We propose a mixture-of-experts (MoE) decoding module, which dynamically routes image features based on the recognized blur degradation, enabling precise and efficient restoration in an end-to-end manner. Our unified approach not only achieves performance comparable to dedicated task-specific models, but also shows promising generalization to unseen blur scenarios, particularly when leveraging appropriate expert selection.

## 1 Introduction

Blur is a fundamental component of the image formation process (Elad & Feuer, 1997; Karaimer & Brown, 2016; Delbracio et al., 2021) and arises during image capture due to factors such as object motion, camera shake, or lens settings like focus and aperture. Conventional approaches (Elad & Feuer, 1997; Levin et al., 2009; Zhu et al., 2012; Schuler et al., 2013) model the blur degradation as:

$$\mathbf{y} = \mathbf{x} \otimes \mathbf{k} + \mathbf{n}, \tag{1}$$

where $\mathbf{y}$ denotes the blurred image, $\mathbf{x}$ is the latent sharp image, $\mathbf{k}$ represents the blur kernel, and $\mathbf{n}$ is the additive noise. The symbol $\otimes$ indicates the convolution operation. This degradation significantly reduces image quality, which may result in user dissatisfaction. Moreover, the space of possible blur kernel $\mathbf{k}$ is infinite, making this inverse problem highly challenging.

To remove blur, numerous methods have been proposed, including conventional blind deconvolution approaches (Levin et al., 2009; Zhu et al., 2012; Schuler et al., 2013), which iteratively optimize a blur kernel $\mathbf{k}$ and the latent sharp image $\mathbf{x}$. In contrast, learning-based methods (Nah et al., 2017; Rim et al., 2020; Kupyn et al., 2018; 2019; Zamir et al., 2022b; Zhou et al., 2022), offer more straightforward approaches. They do not require complex modeling or constraints, as conventional methods do. Instead, deep networks are trained using a deblurring dataset, allowing them to implicitly learn to model degradation and effectively remove it. However, their performance is still limited due to the wide variety of blur patterns in real-world blurred images. For instance, subject movement results in local blur, handheld camera shake can cause global motion blur (Li et al., 2023a; Rim et al., 2020; Nah et al., 2017), and a shallow depth of field may introduce defocus blur.

To address this, most deep learning-based deblurring methods are trained on datasets designed for specific types of blur, such as RealBlur (Rim et al., 2020) for camera shake, ReLoBlur (Li et al., 2023a) for object motion, and DPDD (Abuolaim & Brown, 2020) for defocus blur. As a result, a network trained on defocus blur will fail to resolve motion blur and viceversa. Therefore, there is a clear generalization gap across blur degradations, as shown in Figure 1. This is a clear limitation of current task-specific approaches (Figure 1 (a)).

Recently, *all-in-one image restoration* methods (Li et al., 2022; Potlapalli et al., 2023; Park et al., 2023; Ma et al., 2023; Zhang et al., 2023b;a; Chen et al., 2024; Cui et al., 2024; Lin et al., 2024; Li

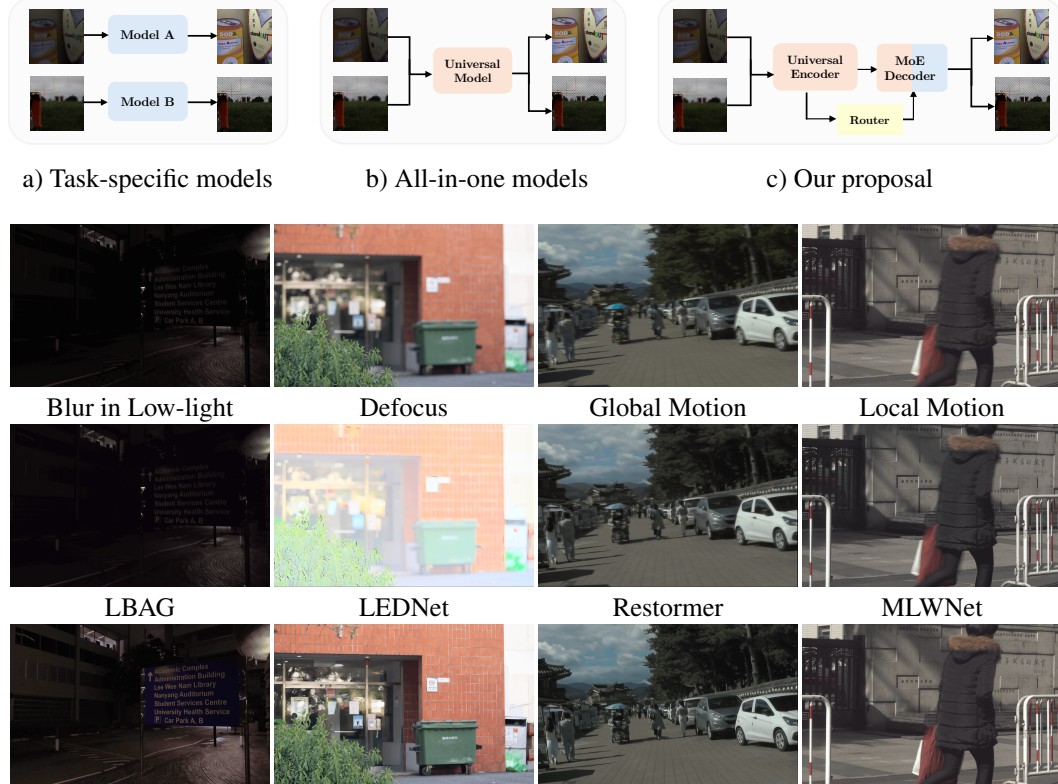

a) Task-specific models      b) All-in-one models      c) Our proposal

Our all-in-one method is the first one able to tackle blurry images under diverse conditions.

Figure 1: **Illustration of different types of blur restoration.** At top, a set of possible deblurring restoration strategies. At bottom, a comparison of task-specific methods and our proposed **DeMoE**. Previous approaches can tackle only specific types of blur and fail to deal with different types of blur. Our all-in-one deblurring method addresses multiple types of blur degradations using a single model. Zoom in for optimal comparison.

et al., 2024) have grown in popularity, establishing themselves as a solid solution to restore images under different conditions using a single neural model *e.g.,* remove noise, correct low illumination, remove haze and rain. Some of these multi-task methods explicitly exploit (or implicitly) the similarities that can be found between different degradations (Cui et al., 2024).

These approaches –illustrated in Figure 1 (b)– inspire us to pose the following questions: *How can we efficiently restore blurry images with diverse types of blur? How similar are the different blur degradations? Can we use a single robust model to restore different types of blur?* Following all-in-one restoration, we aim to simplify the following pipeline into an end-to-end neural network: (i) recognize the degradation *i.e.,* type of blur, (ii) select the degradation-specific (or task-specific) model, (iii) restore the image $y$ using the optimal "expert" model.

**Our contributions.** In this work, we propose the first all-in-one deblurring method to restore efficiently "any" blurry image. Following recent developments in all-in-one restoration, we train our model using diverse datasets that contain multiple blur degradations (global motion, local motion, blur in low-light, and defocus blur). We design a mixture-of-experts (MoE) decoding module to route features according to the detected blur degradations. This novel approach allows us *recognize the type of blur present in the image and restore it*, in an end-to-end manner. Moreover, our method is efficient, robust, and generalizes effectively to out-of-distribution (OOD) real-world blurry images when guided by manual expert selection.

## 2 RELATED WORK

**Image Deblurring.**    Image deblurring aims to recover sharp images from blurred ones caused by camera shake, object movement, and defocus blur. Most existing approaches (Nah et al., 2017; Tao et al., 2018; Zhang et al., 2019; Kupyn et al., 2019) primarily target motion blur and use the GoPro dataset (Nah et al., 2017) as the main benchmark. However, even within motion blur, the types of blur can vary significantly. Rim et al. (2020) presented the RealBlur dataset, comprising real images blurred caused by camera shake. Li et al. (2023a) introduced the ReLoBlur dataset to tackle local motion deblurring, emphasizing moving objects blurred against static backgrounds. Recently, Zhou et al. (2022) introduced LOLBlur, the joint low-light enhancement and deblurring task, where degraded images suffer from both motion blur and low illumination.

Recent defocus deblurring methods have been advanced through dual-pixel imaging. Abuolaim & Brown (2020) established the DPDD dataset to leverage the complementary information from dual-pixel views. Subsequent work includes IFAN (Abuolaim et al., 2022), which synthesizes dual-pixel images via multi-task learning, and Son et al. (2021), who employed kernel-sharing parallel atrous convolutions.

Recent advancements in sophisticated architectures (Cho et al., 2021; Zamir et al., 2022a; 2021; Jiang et al., 2024; Tsai et al., 2022b;a; Kong et al., 2023) have led to further improvements in handling motion and defocus blur. Despite these advancements, each model remains specialized and constrained to particular blur types, necessitating separate training procedures. Our approach is the first to propose a unified all-in-one deblurring model that integrates multiple blur modalities.

**All-in-One Image Restoration.**    Most methods in the literature are designed to tackle a single degradation *e.g.,* noise, blur, low-light, rain. However, in many cases their real-world applications are limited due to the required resources *i.e.,* allocating different task-specific models in memory and selecting the specific model on demand.

In recent years, all-in-one (also known as multi-task) image restoration has emerged as a possible solution to such limitations (Park et al., 2023; Zhang et al., 2023a; Yao et al., 2023; Valanarasu et al., 2022; Cui et al., 2024; Chen et al., 2024). These methods use a single neural network to tackle different degradation types and levels. We can highlight AirNet (Li et al., 2022) and PromptIR (Potlapalli et al., 2023; Conde et al., 2024) as the early proposed solutions. *The image restoration pipeline now becomes an end-to-end neural network, instead of an ensemble of multiple task-specific models.*

These methods use different techniques to learn effective multi-task representations. For instance, a degradation classification model (Li et al., 2022; Park et al., 2023; Lin et al., 2024), and guidance embeddings ("prompts") (Potlapalli et al., 2023; Ma et al., 2023; Conde et al., 2024) that help the model discriminate the different types of degradation in the image. Some works also explore Mixture of Experts (MoEs) (Jacobs et al., 1991; Xu et al., 1994; Shazeer et al., 2017; Guo et al., 2024; Ren et al., 2024), which allows to learn implicitly task-specific experts *within* the neural network.

## 3 METHOD

### 3.1 PRELIMINARIES: DEBLURRING SIMILARITY ANALYSIS

We aim to find a global function $f_\Theta$ able to recover sharp images $x$ from any blurry image $y$, without having any prior information about the blur $\mathbf{k}$ – thus, blind image deblurring.

Let us consider two neural networks with the same architecture, $f_{\Theta 1}$ and $f_{\Theta 2}$, the first is trained to solve global motion deblurring, the second is trained to solve defocus deblurring. We assume the following statement to be true: given $f_{\Theta 1}$ and $f_{\Theta 2}$ trained for similar tasks, the learned parameters $\Theta_1$ and $\Theta_2$ shall be very similar (Nguyen et al., 2020). Therefore, we pose the following question:

> Considering the model $f$, and two sets of parameters $\Theta_1$ and $\Theta_2$ optimized for two different deblurring tasks, how similar are the learned parameters and representations?

Before we develop a multi-task deblurring method, we want to understand which layers of a neural network are the most relevant to restore blurry images, and how similar such networks are. We use

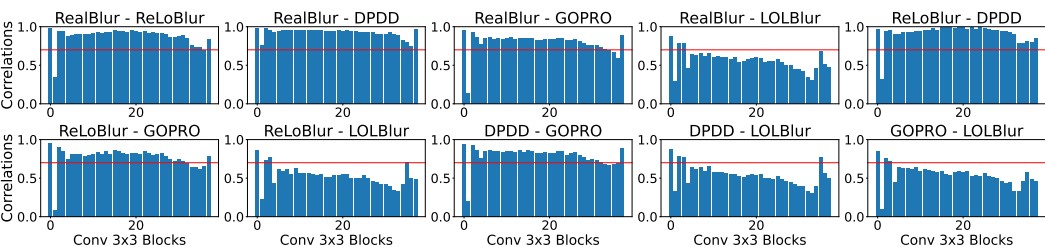

Figure 2: **Network Similarity Study across Deblurring tasks**. We show the layer-wise correlations between different deblurring versions of NAFNet *i.e.,* each bar represents the weights correlation of one layer. Excluding the LOLBlur dataset, all the other weights hold a high-correlation (greater than 0.7 (Asuero et al., 2006), over the red line). For instance, this reveals that a model trained to solve defocus (DPDD) learns similar representations as a model trained for motion deblurring (GOPRO).

NAFNet (Chen et al., 2022) as the baseline model $f$. First, we trained the same model on different blurry-clean datasets, returning a set of specialized weights for each deblurring task $\{\Theta_1, \Theta_2, \dots\}$. Next, we compare the layer-wise Pearson correlations between these model weights. This gives us an intuition on how similar the models behave when restoring different blur degradations.

For this study, we classified the weights defined in NAFNet into the following four types of parameters: simplified channel attention (SCA) blocks, layer normalization, $1 \times 1$ convolutions, and $3 \times 3$ convolutions. Intuitively, the latter operations are the most relevant for image deblurring (Zhu et al., 2012; Schuler et al., 2013) —to restore blurred images, the neighbor pixel information must be combined—, as shown in Figure 2, where the $3 \times 3$ convolution layers hold a high correlation. Also, following the same layer taxonomy, a similarity study between these models using Centered Kernel Alignment (Kornblith et al., 2019), can be found in the supplementary. These results suggest that models $f_{\Theta 1}, f_{\Theta 2}, \dots$ trained for different deblurring tasks (see Figure 2), share common neighbor operations. Therefore, a model trained with different blur datasets will build a similar type of general neighbor operations correction, making it a potential way of restoring general blur.

Based on this study, we propose an All-in-One deblurring method –see Figure 1 (c)–, our model can be divided into: (1) a general feature extractor and classifier encoder, and (2) a MoE decoder. Following the results of the correlation study, the neural blocks used for the method are inspired by the NAFNet blocks. An illustration of the network architecture is shown in Figure 3.

### 3.2 GENERAL BLUR RESTORATION BASELINE

We use NAFNet as our baseline, which employs the Metaformer (Yu et al., 2022) block design and a U-Net (Ronneberger et al., 2015) architecture. The popular NAFBlocks perform two different transformations: channel attention and a feed-forward network (FFN). This efficient architecture represents the state-of-the-art (SOTA) in well-known deblurring datasets, such as GoPro or RealBlur. For this reason and the previous correlation analysis, we use this as our baseline architecture.

As a first improvement, we incorporate an attention-based router $\mathcal{R}$ for degradation classification at the end of the NAFNet encoder, as shown in Figure 3. The encoder $\mathcal{E}$ extracts high and low-frequency features and identifies image degradations with the help of an MLP (multi-layer perceptron) branch. The router output is a vector of normalized weights $\mathbf{w} \in \mathbb{R}^N$, where $N$ is the number of experts (Jacobs et al., 1991; Xu et al., 1994). We use $N = 5$ in DeMoE, since we use five different deblurring datasets. For the considered training datasets, the router achieves **perfect classification** accuracy *i.e.,* $> 95\%$ accuracy. More explained details can be found in the supplementary material.

### 3.3 MIXTURE OF DEBLURRING EXPERTS

Based on the similarity study (Sec. 3.1), we aim to learn (within a single all-in-one model) the task-specific features that account for the notable model differences *i.e.,* attention and FFN weights.

After extracting features and recognizing the blur degradation, the MoE decoder restores the image. In each level of the decoder, we add a MoEBlock that incorporates $N$ different experts e tailored to

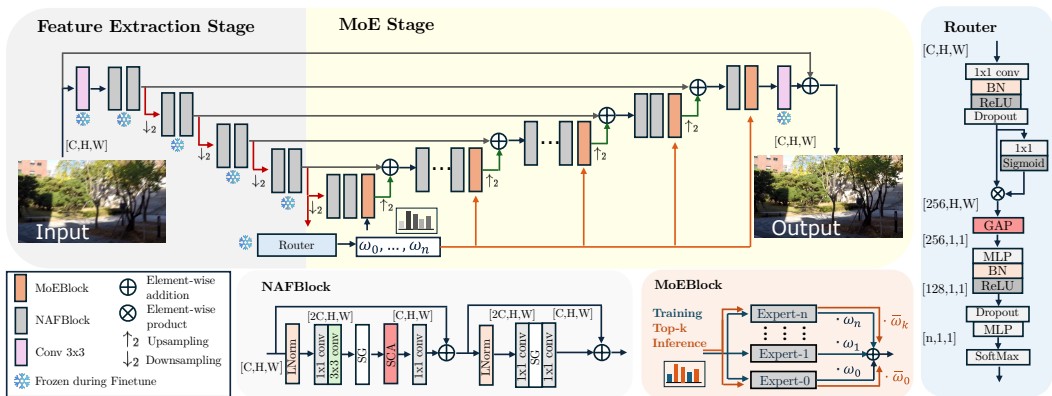

Figure 3: **DeMoE Network Architecture**. We adopt NAFNet as backbone. The encoder generates a feature space optimized for both restoration and degradation classification. The router uses the encoded features to determine the degradation and assigns weights to each expert. During training, all experts contribute to restoration; during inference, only the top-k experts—selected based on router weights—are used to produce the final output.

restore specific blur degradations. During training, we use a differentiable softmax $\sigma$ gating, such that all experts are used for restoration, even those with low-weight contributions (Yu et al., 2021; Guo et al., 2024) – see Eq 2. During inference, only the top-k experts are selected based on the $k$ larger weights given by the router $\sigma$ gating, considering the contribution of only the most relevant experts to restore the image. The output result of the MoEBlock follows the relation:

$$\hat{\mathbf{h}} = \sum_{i=0}^{N} \mathbf{w}_i \cdot \mathrm{e}_i(\mathbf{h}) \quad \text{where} \quad \mathbf{w} = \sigma(\mathcal{R}(\mathcal{E}(\mathbf{y}))) \ . \tag{2}$$

The feature map $\mathbf{h}$ passes through each of the selected experts $\mathrm{e}_i$, returning a restored feature map $\hat{\mathbf{h}}$ – see Figure 3 MoEBlock. Finally, at the end of each MoEBlock, all restored feature maps are added together weighted according to the relevance given by the router $\mathbf{w}_i$. Note that the same router weights are applied in all MoEBlocks. Note that when we use $k = 1$ (top-1) expert during inference, the DeMoE architecture is equivalent to our NAFNet baseline, saving computation and runtime.

### 3.4 DEGRADATION-AWARE PRE-TRAINING

The network is trained end-to-end using a multi-step approach similar to Lin et al. (2024); Chen et al. (2024); Hu et al. (2025). Moreover, we use a combination of image regression and classification losses. These losses are defined as $L_{pixel} = ||\mathbf{x} - \hat{\mathbf{x}}||_1$ and $L_{class} = -\sum p(x) \log \mathbf{w}$, where $L_{pixel}$ is the L1 loss calculated using $f(\mathbf{y}) = \hat{\mathbf{x}}$ as the enhanced image and $\mathbf{x}$ as the ground-truth. On the other hand, $L_{class}$ is the Cross-Entropy loss. The distributions compared in this case are the ground-truth degradation label of each of the images $p(x)$ and the predicted weights by the router $\mathbf{w}$.

The network is trained end-to-end in a two-step pipeline. First, we train the baseline and router using the following combination of losses $\mathcal{L} = \lambda_p \cdot L_{pixel} + \lambda_{cl} \cdot L_{class}$. The constants $\lambda_p$ and $\lambda_{cl}$ are, respectively, 1.0 and 0.001. Then, we freeze the router and encoder layers and finetune the MoEBlocks and decoder layers using only $\mathcal{L} = L_{pixel}$.

## 4 EXPERIMENTAL RESULTS

**The AIO-Blur Dataset**   To train and evaluate all-in-one deblurring methods, a dataset containing various types of blur degradation is required. However, there is no such dataset so far. To address this, we construct the **A**ll-**I**n-**O**ne-**Blur** (AIO-Blur) dataset by collecting diverse types of blur datasets from open-source repositories. Specifically, AIO-Blur comprises a diverse set of blur types, including: camera motion blur (*i.e.,* RealBlur), object motion blur (*i.e.,* ReLoBlur), camera and object motion blur (*i.e.,* GoPro), low-light and motion blur (*i.e.,* LOLBlur, and defocus blur (*i.e.,* DPDD). We follow the original train/test splits for each dataset.

| Dataset | Train | Train/Eval Split | Real-world | Blur type |
|---------|-------|------------------|------------|-----------|
| RealBlur | ✓ | 3758 / 980 | ✓ | Camera Motion |
| ReLoBlur | ✓ | 2010 / 395 | ✓ | Local Motion |
| LOLBlur | ✓ | 10200 / 1800 | ✗ | Low-Light Motion |
| GoPro | ✓ | 2103 / 1111 | ✗ | Camera/Object Motion |
| DPDD | ✓ | 350 / 75 | ✓ | Defocus |
| Real-LOLBlur | ✗ | - / 872 | ✓ | Low-Light Motion |
| RealDOF | ✗ | - / 50 | ✓ | Defocus |
| RSBlur | ✗ | 8878 / 3360 | ✓ | Camera/Object Motion |

Figure 4: (Left) t-SNE distribution of the testing images in the AIO-Blur and AIO-Blur-OOD datasets. (Right) Main specifications of each dataset used for training and testing. We use three real-world datasets to test the robustness of the models in out of distribution (OOD) scenarios.

Table 1: **All-in-one Single Image Deblurring**. We compare *state-of-the-art* image restoration methods trained for "all-in-one deblurring". Each method is tested on the most representative benchmarks for **motion deblurring** (RealBlur, GoPro), **low-light deblurring** (LOLBlur), **local motion deblurring** (ReLoBlur), and **defocus deblurring** (DPDD). We report PSNR↑ (dB) / SSIM↑ / LPIPS↑ (Zhang et al., 2018) across datasets. We also report computational cost based on parameters (M) / MACs (G) / runtime (ms). MACs and Runtime were calculated using crops of size 256px x 256px. Runtime was averaging the forward pass of 1000 iterations using an NVIDIA RTX 4090. The **bold** and underlined stand for the best and second best results, respectively. Our method, DeMoE, has the best performance in general image deblurring and is the second one in efficiency.

| Method | Computational Cost | RealBlur | ReLoBlur | DPDD | LOLBlur | GoPro | Average |
|--------|--------------------|----------|----------|------|---------|-------|---------|
| PromptIR | 35.59 / 158.4 / 34.4 | 29.23 / 0.867 / 0.198 | 34.50 / 0.925 / 0.189 | 25.21 / 0.766 / 0.282 | 26.03 / 0.846 / 0.206 | 28.18 / 0.849 / 0.213 | 28.63 / 0.851 / 0.218 |
| Restormer | 26.13 / 141.24 / 33.2 | 28.80 / 0.866 / 0.217 | 34.33 / 0.924 / 0.205 | 24.72 / 0.764 / 0.309 | 25.61 / 0.844 / 0.227 | 27.76 / 0.848 / 0.233 | 28.24 / 0.853 / 0.238 |
| FFTFormer | 16.56 / 131.75 / 61.8 | 28.08 / 0.839 / 0.230 | 34.45 / 0.922 / 0.189 | 25.04 / 0.776 / 0.281 | 22.72 / 0.825 / 0.246 | 27.61 / 0.841 / 0.226 | 27.58 / 0.841 / 0.234 |
| SFHFormer | **7.67** / 52.32 / 32.5 | **29.69** / **0.888** / **0.172** | 34.32 / 0.924 / **0.188** | 25.15 / 0.779 / 0.288 | 25.61 / 0.857 / 0.211 | 29.09 / 0.884 / 0.194 | 28.77 / 0.867 / 0.211 |
| NAFNet | 10.08 / **11.02** / **8** | 29.18 / 0.883 / 0.203 | **34.56** / **0.926** / 0.228 | **25.60** / 0.795 / 0.266 | 26.69 / 0.871 / 0.188 | 29.56 / **0.890** / 0.191 | 29.12 / **0.873** / 0.215 |
| **DeMoE**$_{k=1}$ | 20.15 / 11.05 / 13.1 | 28.96 / 0.884 / 0.198 | 34.52 / 0.925 / 0.232 | 25.56 / **0.797** / 0.258 | **26.84** / **0.878** / 0.175 | **30.06** / **0.900** / 0.176 | **29.19** / **0.877** / **0.208** |

Additionally, we construct **test-only datasets** to evaluate the robustness of methods on out-of-distribution (OOD) data. The dataset includes test sets of Real-LOLBlur (Zhou et al., 2022), RealDOF (Lee et al., 2021), and RSBlur. For Real-LOLBlur, we omit its RealBlur subset, since RealBlur is already included in AIO-Blur. We refer to this dataset as AIO-Blur-OOD.

Figure 4 provides an overview of AIO-Blur and AIO-Blur-OOD datasets, along with a t-SNE (Van der Maaten & Hinton, 2008) visualization of cascade CLIP (Radford et al., 2021) blurry features for each dataset. The clear separation among datasets highlights the diversity of blur types across various environments. This motivates the need for an all-in-one deblurring model able to handle such diverse degradations. A more detailed explanation on the datasets can be found in the supplementary material, as well as extensive implementation details and additional results.

## 4.1 RESULTS ON AIO-BLUR

In this section, we evaluate DeMoE on the AIO-Blur dataset and compare it with other deblurring and all-in-one methods, including Restormer (Zamir et al., 2022a), PromptIR (Potlapalli et al., 2023), FFTFormer (Kong et al., 2023), SFHFormer (Jiang et al., 2024), and NAFNet (Chen et al., 2022). We use PromptIR as the canonical AIO method for comparison. For fairness, all methods are trained on the AIO-Blur dataset. Table 1 presents quantitative results on test sets, where DeMoE (based on NAFNet) is the best method overall. SFHFormer achieves the best performance on RealBlur, which we attribute to its Fourier-domain branch being particularly effective for camera motion blur. Table 1 also presents a computational cost analysis, including the number of parameters, runtime, and Multiply-Accumulate Operations (MACs). As shown in the table, DeMoE is the second most efficient method regarding both runtime and MACs. Compared to PromptIR, our method uses ≈ 2× fewer parameters, ≈ 15× fewer operations, and is 2× faster.

Table 2: **Quantitative evaluations on AIO-Blur-OOD.** DeMoE$^\dagger_{k=1}$ denotes using a manually selected expert. For reference, we also report the state-of-the-art for each dataset.

| | Method | RealDOF | RSBlur | Real-LOLBlur |
|---|---|---|---|---|
| | | PSNR↑ / SSIM↑ / LPIPS↓ | PSNR↑ / SSIM↑ | MUSIQ↑ / NRQM↑ / NIQE↓ |
| General deblur | Restormer | 22.98 / 0.684 / 0.471 | 14.04 / 0.556 | 27.71 / 5.169 / 7.381 |
| | FFTFormer | 23.36 / 0.697 / 0.440 | 15.60 / 0.589 | 30.18 / 5.780 / 6.469 |
| | SFHFormer | 24.20 / 0.736 / 0.428 | 13.54 / 0.553 | 29.58 / 5.320 / 6.806 |
| | NAFNet | 24.64 / **0.762** / 0.382 | 15.00 / 0.576 | 32.02 / 5.515 / 6.865 |
| | **DeMoE**$_{k=1}$ | 24.59 / 0.758 / 0.377 | 19.23 / 0.640 | 31.34 / 5.470 / 7.560 |
| | **DeMoE**$^\dagger_{k=1}$ | 24.59 / 0.758 / **0.377** | 27.53 / 0.763 | 38.91 / **6.113** / 5.820 |
| Task-specific | IFAN | **24.71** / 0.748 / **0.306** | - | - |
| | MLWNet-B | - | **30.91** / **0.818** | - |
| | LEDNet | - | - | **39.11** / 5.643 / **4.764** |

Table 3: **Quantitative comparisons with task-specific methods.** DeMoE is the proposed all-in-one deblurring method trained on the AIO-Blur dataset, while the other methods are task-specific and trained on their respective datasets. DeMoE$^*_{k=1}$ denotes the task-specific DeMoE (equivalent to fine-tuned NAFNet) trained on each respective dataset.

| Method | PSNR↑ | SSIM↑ | LPIPS↓ |
|---|---|---|---|
| KinD++ (Zhang et al., 2021) | 21.26 | 0.753 | 0.359 |
| DRBN (Yang et al., 2020) | 21.78 | 0.768 | 0.325 |
| DeblurGAN-v2 (Kupyn et al., 2019) | 22.30 | 0.745 | 0.356 |
| DMPHN (Zhang et al., 2019) | 22.20 | 0.817 | 0.301 |
| MIMO (Cho et al., 2021) | 22.41 | 0.835 | 0.262 |
| LEDNet (Zhou et al., 2022) | 25.74 | 0.850 | 0.224 |
| DarkIR (Feijoo et al., 2025) | **27.00** | **0.883** | **0.162** |
| **DeMoE**$^*_{k=1}$ | 26.94 | 0.881 | 0.172 |
| **DeMoE**$_{k=1}$ | 26.84 | 0.878 | 0.175 |

**LOLBlur** results

| Method | PSNR↑ | SSIM↑ |
|---|---|---|
| DeepDeblur (Nah et al., 2017) | 33.05 | 0.8946 |
| DeblurGAN-v2 (Kupyn et al., 2019) | 33.85 | 0.9027 |
| SRN-DeblurNet (Tao et al., 2018) | 34.30 | 0.9238 |
| HINet (Chen et al., 2021) | 34.36 | 0.9151 |
| MIMO-Unet (Cho et al., 2021) | 34.52 | **0.9250** |
| LBAG (Li et al., 2023a) | **34.66** | 0.9249 |
| **DeMoE**$^*_{k=1}$ | 34.50 | **0.926** |
| **DeMoE**$_{k=1}$ | 34.52 | 0.925 |

**ReLoBlur** results

| Method | PSNR↑ | SSIM↑ | LPIPS↓ |
|---|---|---|---|
| DMENet (Lee et al., 2019) | 23.41 | 0.714 | 0.349 |
| EBDB (Karaali & Jung, 2017) | 23.45 | 0.683 | 0.336 |
| JNB (Shi et al., 2015) | 23.84 | 0.715 | 0.315 |
| DPDNet (D) (Abuolaim & Brown, 2020) | 25.13 | 0.786 | 0.223 |
| KPAC (Son et al., 2021) | 25.22 | 0.774 | 0.227 |
| IFAN (Lee et al., 2021) | 25.37 | 0.789 | 0.217 |
| Restormer (Zamir et al., 2022a) | **25.98** | **0.811** | **0.178** |
| **DeMoE**$^*_{k=1}$ | 25.67 | 0.800 | 0.255 |
| **DeMoE**$_{k=1}$ | 25.56 | 0.797 | 0.258 |

**DPDD** results

| Method | PSNR↑ | SSIM↓ |
|---|---|---|
| DeblurGAN-v2 (Kupyn et al., 2019) | 29.69 | 0.870 |
| MPRNet (Zamir et al., 2021) | 31.76 | 0.922 |
| MIMO-UNet+ (Cho et al., 2021) | 32.05 | 0.921 |
| BANet (Tsai et al., 2022b) | 32.42 | 0.929 |
| Stripformer (Tsai et al., 2022a) | 32.48 | 0.929 |
| FFTformer (Kong et al., 2023) | 32.62 | 0.933 |
| GRL-B (Li et al., 2023b) | 32.82 | 0.932 |
| MLWNet-B (Gao et al., 2024) | **33.84** | **0.941** |
| **DeMoE**$^*_{k=1}$ | 29.08 | 0.886 |
| **DeMoE**$_{k=1}$ | 28.96 | 0.884 |

**RealBlur-J** results

| Method | PSNR↑ | SSIM↑ |
|---|---|---|
| DeblurGAN-v2 (Kupyn et al., 2019) | 29.55 | 0.934 |
| DeepDeblur (MSCNN) (Nah et al., 2017) | 29.08 | 0.914 |
| MPRNet (Zamir et al., 2021) | 32.66 | 0.959 |
| Restormer (Zamir et al., 2022a) | 33.57 | 0.966 |
| MLWNet-B (gao2024efficient) | 33.83 | 0.968 |
| FFTformer (Kong et al., 2023) | **34.21** | **0.969** |
| **DeMoE**$^*_{k=1}$ | 30.17 | 0.901 |
| **DeMoE**$_{k=1}$ | 30.06 | 0.900 |

**GoPro** results

## 4.2 RESULTS ON AIO-BLUR-OOD

We evaluate DeMoE on the AIO-Blur-OOD dataset and report the quantitative results in Table 2, along with the task-specific baselines —IFAN (Lee et al., 2021), MLWNet-B (Gao et al., 2024) and LEDNet (Zhou et al., 2022)— for reference. While all methods perform comparably to the task-specific model on RealDOF, performance drops significantly on RSBlur and Real-LOLBlur. We suspect that the reason is that the distributions of RSBlur and Real-LOLBlur are significantly different from those of AIO-Blur, as shown in Figure 4. As a result, all methods struggle to deblur these datasets. In the case of DeMoE, the router classifier fails to correctly select an expert on these datasets –see Section E–, ultimately leading to sub-optimal performance. However, thanks to its design, *expert selection can be manually controlled by users* when necessary. Thus, for RealDOF, RSBlur and Real-LOLBlur, we manually assign the experts of DPDD, RealBlur and LOLBlur, respectively. Table 2 shows the DeMoE results with manual expert selection, achieving superior performance across all datasets. Even when the distribution of the test data is shifted, DeMoE can easily handle the shift through manual expert selection, whereas other methods cannot.

## 4.3 COMPARISON WITH TASK-SPECIFIC METHODS

We compare DeMoE, the first all-in-one deblur model, with task-specific methods that are carefully designed for specific types of blur. We also evaluate *task-specific versions of DeMoE$^*$*, where we use a single expert in the decoder and we fine-tune in each specific dataset. This variant of DeMoE can be interpreted as the *upper-bound limit of each DeMoE expert* and is equivalent to a single specialized NAFNet model – see DeMoE$^*$ in Table 3.

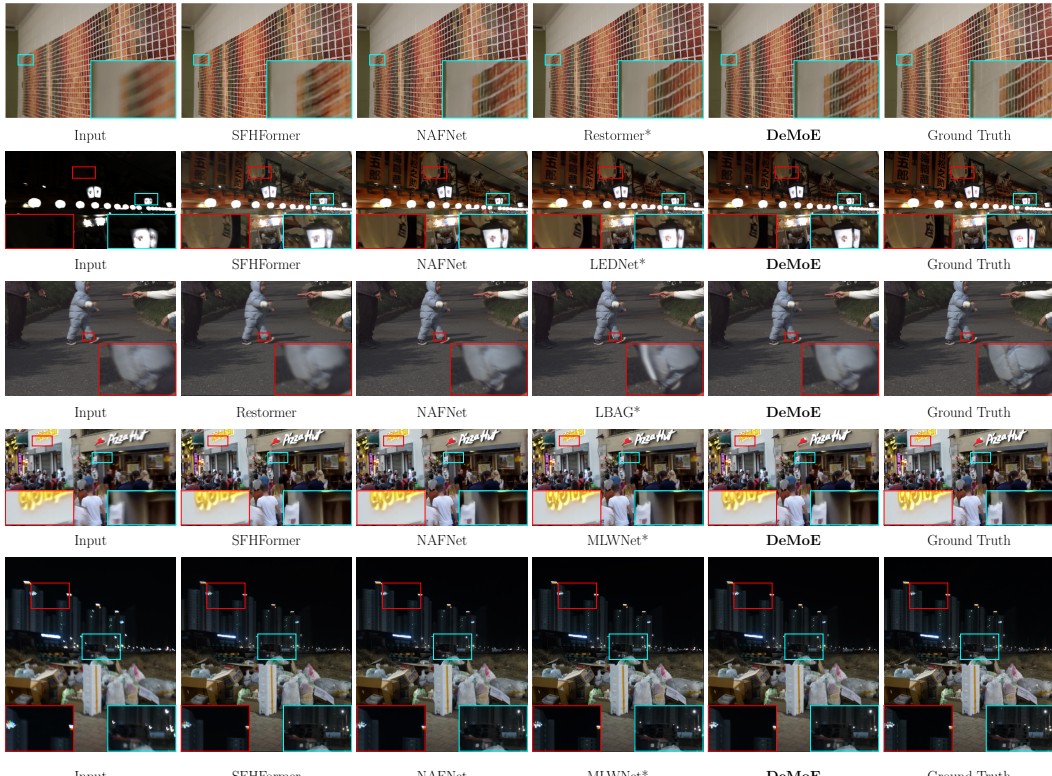

Figure 5: **Qualitative comparison of the general deblur methods.** Methods with * are SOTA task-specific methods. Results, from top to bottom, of the following datasets: DPDD, LOLBlur, ReLoBlur, GoPro and RealBlur. Our method, DeMoE, provides results comparable to SOTA.

All-in-one (AIO) methods for general image restoration typically underperform compared to task-specific methods. For instance, AIO Restormer (Zamir et al., 2022b; Conde et al., 2024; Zhang et al., 2023b) achieves a PSNR (dB) of 27.22 in GoPro deblurring and 20.41 in LOL correction, while the task-specific designed methods FFTformer (Kong et al., 2023) and LLFormer (Wang et al., 2023) achieve 34.21 and 23.64, respectively.

**Motion Deblurring**   Table 3 shows results on the RealBlur-J dataset (Rim et al., 2020), our model achieves competitive performance, closely matching the specialized DeblurGAN-v2 (Kupyn et al., 2019), yet without dedicated training, and being more versatile and efficient.

**Local Motion Deblurring**   In the local motion deblurring scenario (Table 3), our unified multi-task approach attains results (34.52 dB PSNR, 0.925 SSIM) comparable to LBAG (Li et al., 2023a), the current SOTA method specifically designed for local blur (34.66 dB PSNR, 0.925 SSIM).

**Defocus Deblurring**   Table 3 illustrates the results for defocus deblurring on the DPDD dataset. While specialized single-image defocus methods such as Restormer achieve slightly higher metrics (25.98 dB PSNR, 0.811 SSIM), our unified model still achieves competitive performance (25.56 dB PSNR, 0.797 SSIM), closely trailing the best specialized single-task methods. Considering our model handles multiple blur types simultaneously, this marginal performance gap is outweighed by the significant efficiency and flexibility benefits provided by our multi-task design.

**Low-Light Deblurring**   In low-light deblurring (Table 3), our model delivers excellent results (26.84 dB PSNR, 0.878 SSIM, 0.175 LPIPS), clearly exceeding the task-specific LEDNet (25.74 dB PSNR, 0.850 SSIM, 0.224 LPIPS). The strong performance in this particularly challenging low-light scenario further illustrates our model's adaptability and underscores its practical efficiency, as it alleviates the need for separate task-specific training and architectures.

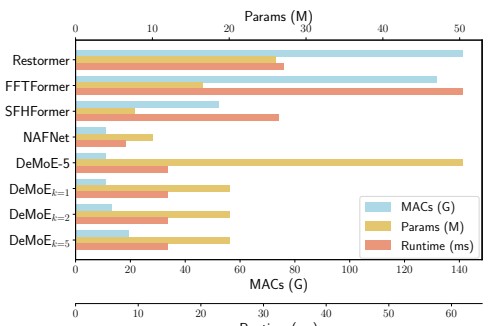

Figure 6: (Left) Computation cost representation in parameters, MACs and runtime of some general deblur methods considered in Table 1. (Right) Ablation study performed on DeMoE. The first column of the table states the configuration used during training. The values on PSNR, SSIM and LPIPS are referred to the averaged metrics of the AIO-Blur dataset.

**Upper-bound of DeMoE**    Table 3 shows the performance of task-specific DeMoE*. The average metrics of all task-specific DeMoE* are 29.27 PSNR and 0.879 SSIM, thus, the DeMoE performance drop is just $\approx 0.3\%$. This demonstrates that each expert in DeMoE effectively learns to deblur specific blur types and achieves performance comparable to task-specific models.

**Qualitative Results**    In Figure 5, qualitative samples are presented. We compare DeMoE with general deblur methods trained in the AOI-Blur dataset and the SOTA result in that specific dataset. We provide more qualitative results, including the AOI-Blur-OOD dataset, in the supplementary.

## 5 DISCUSSION

**Ablation Study**    In Table 6 (right) we present a study of different training approaches for DeMoE. As in some previous works (Lin et al., 2024; Hu et al., 2025), pre-training and expert MoE fine-tuning is optimal. In Section F extended ablation studies support the proposed architecture.

**Efficiency Study**    Figure 6 (left) illustrates the computational cost of deblurring methods, where DeMoE achieves a notable reduction in MACs and runtime compared to models such as FFTFormer and Restormer. Note that DeMoE$_{k=1}$ is as efficient as NAFNet – the additional parameters and operations are due to the router and do not affect the runtime. The figure also presents DeMoE-5, an ensemble comprising the five task-specific DeMoE* networks in Table 3 – similar to an ensemble of five task-specific NAFNets. Compared to the ensemble, DeMoE reduces the parameter count from 50.4 M to 20.15 M ($2.5\times$ fewer), with only a 0.3% drop in PSNR and SSIM.

**Performance of DeMoE**    DeMoE sometimes performs worse than task-specific methods, particularly on RealBlur and GoPro. We suspect that this is due to the inherent limitations of general all-in-one methods, and due to our limited number of parameters to maintain efficiency.

**Limitations and Future Work**    As discussed in Section 4.2, one limitation of DeMoE is the router degradation classifier. If the router cannot identify properly the degradation, the experts are not properly used, leading to sub-optimal results. This issue could be mitigated by *enriching the training dataset* with a broader range of images and blur, enabling to generalize over more diverse scenes.

## 6 CONCLUSION

We introduce the first all-in-one deblurring method capable of efficiently restoring images with diverse blur degradations such as motion blur, blur in low-light conditions, and defocus blur. We employ a mixture-of-experts (MoE) decoding module, which dynamically routes features based on the recognized blur degradation, enabling precise and efficient restoration in an end-to-end manner. Our unified approach, as a single model, achieves performance comparable to dedicated task-specific models across five datasets, yet being general and more robust. Our method will be open-source.

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

# SUPPLEMENTARY MATERIAL

## A SIMILARITY WEIGHTS ANALYSIS

**Correlation Analysis** During the preliminary step of this research, a study of weight similarities was developed. Given each of the datasets that form part of AIO-Blur, a baseline NAFNet (Chen et al., 2022) $f_\Theta$ has been trained on each of them. Then, pairs of training weights $\Theta_1$, $\Theta_2$ of the same architecture $f_\Theta$ in different datasets are compared using two similarity methods: the Pearson correlation and CKA (Kornblith et al., 2019). In Figures 7, 8 and 9 the results of this study are shown.

To calculate the correlations, we classify the different blocks of the NAFNet architecture into four types of layers: pixel-wise $1 \times 1$ convolutions, $3 \times 3$ convolutions, layer normalization, and simplified channel attention (SCA) layers. Each of these layers is composed of a set of $C$ filters that resemble the channel size of the features that are introduced to the layer. For each pair of weights $\Theta_1$, $\Theta_2$, we compute the correlation of their layer values per filter. Then, the mean of these filter correlations is calculated to define the final correlation value of the layer. The correlation for each filter is given by

$$r = \frac{\sum_i (\Theta_{1,i} - \bar{\Theta}_1)(\Theta_{2,i} - \bar{\Theta}_2)}{\sqrt{\sum_i (\Theta_{1,i} - \bar{\Theta}_1)^2 \sum (\Theta_{2,i} - \bar{\Theta}_2)^2}}, \tag{3}$$

where $\bar{\Theta}_j$ and $\Theta_{j,i}$ are the average filter values and an element value of a filter for $\Theta_j$ task-specific weights, respectively. Finally, we calculate the mean of the correlations of the filters in order to get the mean correlation for each block. The mean is calculated following

$$R = \frac{\sum_{i=1}^C r_i}{C}, \tag{4}$$

where $C$ is the number of filters in the layer and $r_i$ is the correlation of the $i$th filter.

Figure 7 gives us a general idea on how these weights are related and that in general, LOLBlur (Zhou et al., 2022) has a lower correlation with the other datasets, which can be related not only to blur degradations, but also to low-light ones. We also observe that convolution blocks with a kernel size of 3 exhibit a strong correlation across all dataset pairs. This is likely because these blocks uniquely incorporate information from neighboring pixels in their computations, making them particularly well-suited for capturing the local structure involved in convolutional operations such as blurring. The results in Figure 7 suggest that the impact of these blocks on blur restoration is consistent across different cases, regardless of the specific type of blur degradation.

We confirmed that these results were consistent with our interpretation of the Pearson correlation values by computing the weight correlations between the models trained on each deblurring task and those trained on a low-light restoration task. The selected training dataset for the low-light task weights was LOLv2-real (Yang et al., 2021). The results of this study are shown in Figure 8, where it can be seen that neither of the layers considered shows any correlation with the AIO-Blur dataset weights. This supports the idea that weights $\Theta_1$, $\Theta_2$ trained in different blur degradations share similarities.

**Centered Kernel Alignment** Following the same strategy to calculate correlations per filter, we computed the Centered Kernel Alignment similarity index by (Kornblith et al., 2019). Since Pearson's coefficient does not capture non-linear relationships, we also performed a CKA analysis to ensure the results obtained in the previous study. The authors introduced two variants of the CKA by changing the kernel function: a linear kernel or an RBF (radial basis function) kernel. We chose to use the second one in order to capture non-linear relationships. The values range from 0 to 1, with higher values indicating greater similarity between the weights.

Let $X \in \mathbb{R}^{n_1 \times p}$ and $Y \in \mathbb{R}^{n_2 \times p}$ denote two matrices of activations of $p$ neurons for $n_1$ and $n_2$ examples, respectively; $K_{ij} = k(\mathbf{x_i}, \mathbf{x_j})$ and $L_{ij} = l(\mathbf{y_i}, \mathbf{y_j})$, with $k$ and $l$ two different kernels. The CKA is calculated following

$$CKA(K, L) = \frac{HSIC(K, L)}{\sqrt{HSIC(K, K) * HSIC(L, L)}}, \tag{5}$$

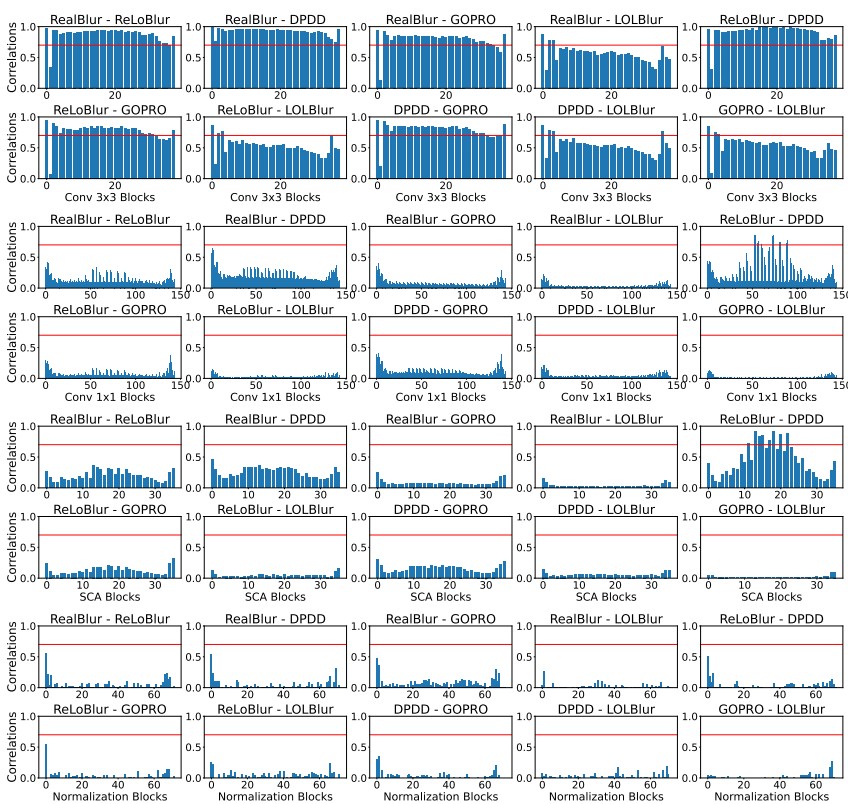

Figure 7: Correlations calculated for different types of neural layers in NAFNet between task-specific weights. The layer considered for each diagram is stated in x axis. Correlation values over 0.7 state a high-correlation (Asuero et al., 2006).

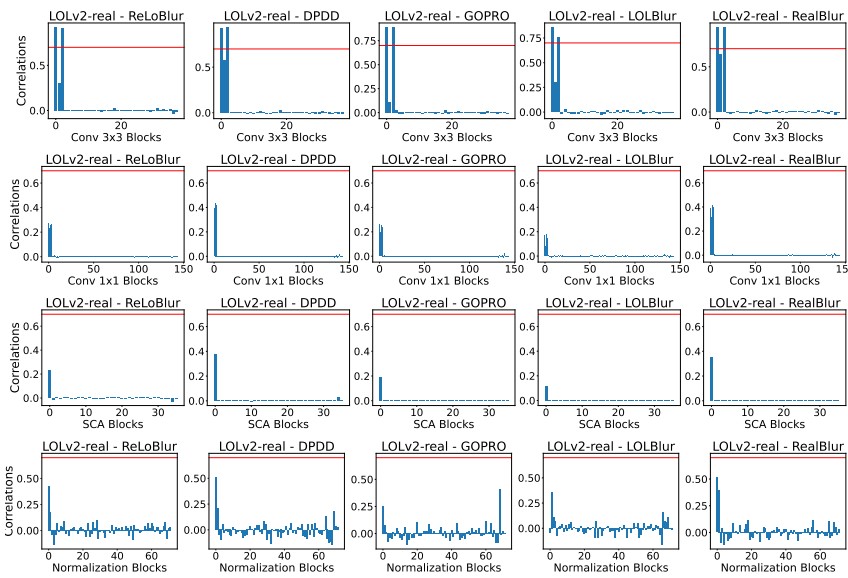

Figure 8: Correlations of the weights compared to a different task.

where $HSIC(K, L)$ represents the Hilbert-Schmidt Independence Criterion empirical estimator (Gretton et al., 2005). This estimator is calculated by

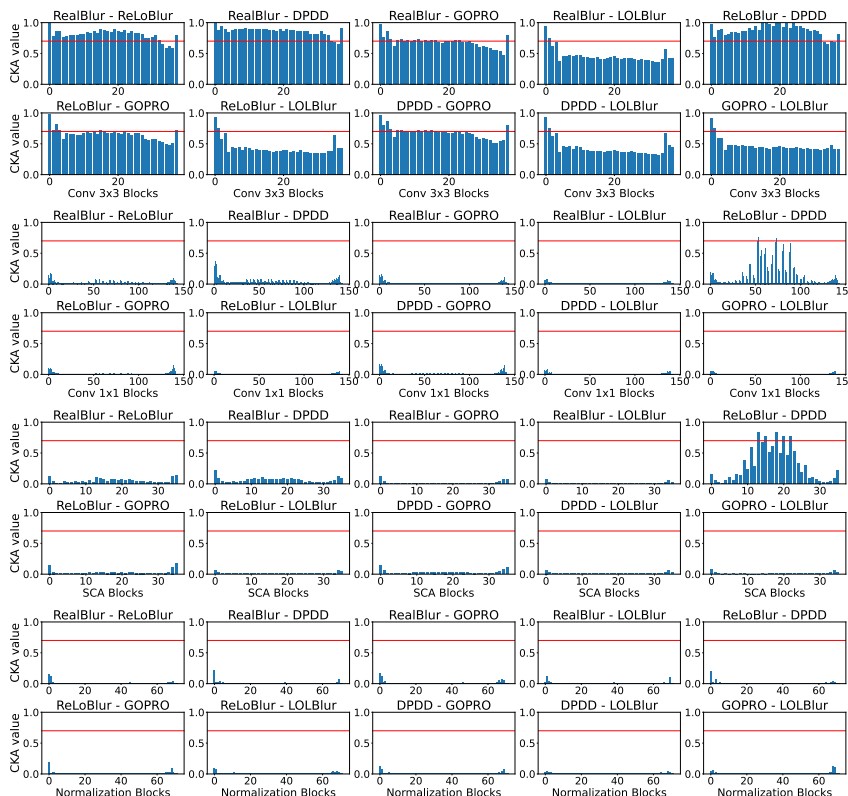

Figure 9: CKA similarity index calculated for different types of neural layers in NAFNet between task-specific weights. Values near 1 represent higher similarity values.

$$HSIC(K, L) = \frac{1}{(n-1)} tr(KHLH), \tag{6}$$

where $H$ is the centering matrix $H_n = I_n - \frac{1}{n} \mathbf{1}\mathbf{1}^T$. We observed that the CKA results are similar to the ones obtained using correlation, as shown in Figure 9. This reinforces the statement on $\Theta_1$ and $\Theta_2$ weights similarities for different deblurring degradations.

**Conclusions**   From this analysis, we can conclude that the operations needed for the deblurring task are similar for different types of blur, based on:

- There is a high correlation between the layer weights in the neighbor-based operations: the kernel size 3 convolutions. These layers have a similar behavior for any type of blur.
- The weighs of the layers that do not operate on neighbors show no significant correlation, thus, there is no similar behavior between these layers.
- CKA analysis shows great similarity indexes between the weights of the layers needed for the deblurring task, which means that they not only have a similar behavior, but they also learn similar representations.

## B   MORE DETAILS OF AIO-BLUR

In this section, we provide more details of the datasets used to construct AIO-Blur.

**GoPro (Nah et al., 2017)** is a *synthetic* dataset for motion blur. Blurred images are generated by averaging consecutive frames from high-speed videos. Then, the center frame of each sequence is used as the ground-truth sharp image. The dataset consists of 2,103 training blur-sharp pairs and 1,111 test pairs.

| Dataset | Original Sample | Final Sample |
|---|---|---|
| RealBlur (Rim et al., 2020) | 3758 | 3758 |
| ReLoBlur (Li et al., 2023a) | 2010 | 4018 |
| LOLBlur (Zhou et al., 2022) | 10200 | 4200 |
| GoPro (Nah et al., 2017) | 2103 | 4206 |
| DPDD (Abuolaim & Brown, 2020) | 350 | 3850 |
| AIO-Blur | **18421** | **20032** |

Table 4: Final distribution of images in the training set of AIO-Blur. The testing sets were not modified.

**RealBlur (Rim et al., 2020)** is a *real-world* dataset for camera motion blur. RealBlur was collected using a dual-camera system, where one camera captures a sharp image with a short exposure time, and the other captures a blurred image with a long exposure time. Using the dual-camera system, RealBlur provides real blurred images caused by camera motion and the corresponding ground-truth sharp images. We used the *RealBlur-J* subset, which provides JPEG RGB images. The dataset contains 3,758 training pairs and 980 testing pairs.

**ReLoBlur (Li et al., 2023a)** is a *real-world* dataset for local motion blur. It was introduced for the task of local motion deblurring, with an emphasis on blurred moving objects against static backgrounds. The dataset was collected by capturing moving objects in front of static backgrounds, using a dual-camera system. It consists of 2,010 training blur-sharp pairs and 395 test pairs.

**LOLBlur (Zhou et al., 2022)** is a *synthetic* dataset for low-light motion blur. Blur typically occurs in low-light environments, such as dimly lit indoor scenes or nighttime, where captured images often suffer not only from motion blur but also from low-light degradation. To address this issue, the dataset was introduced for the joint task of low-light enhancement and deblurring. Blurred images are generated by averaging consecutive frames, and low-light degradation is simulated using EC-Zero-DCE (a variant of Zero-DCE Guo et al. (2020)). LOLBlur consists of 10,200 low-light blurry training pairs and 1,800 testing pairs.

**DPDD (Abuolaim & Brown, 2020)** is a *real-world* dataset for defocus blur. The dataset was collected on static scenes using a single camera mounted on a tripod. A blurred image was captured with a small aperture, and the ground-truth sharp image was captured immediately afterward using a large aperture. DPDD consists of 500 defocus-sharp pairs, split into training (70%), validation (15%) and testing (15%) sets. For AIO-Blur, only the training and testing sets are used.

In addition, we constructed another deblurring dataset, namely **AIO-Blur-OOD**, to evaluate the robustness of methods on out-of-distribution (OOD) data. The dataset is composed of the following open-source datasets:

**RealDOF (Lee et al., 2021)** is a *real-world* dataset for defocus blur. RealDOF was collected using a dual-camera setup, where one camera captures all-in-focus images with a small aperture, and the other captures defocused images with a large aperture. The dataset is test-only dataset consisting 50 scenes.

**RSBlur (Rim et al., 2022)** is a *real-world* dataset for motion blur. The dataset is composed of a total of 13,358 real blurred images of 697 scenes. This pairs are split into 8,878 training, 1,120 validation, and 3,360 test sets. The dataset was collected using a dual-camera system including both camera shake and object motion blur.

**Real-LOLBlur (Zhou et al., 2022)** is a *real-world* dataset for low-light and motion blur. The dataset contains 872 real-world low-light blurry images without ground-truth sharp images. For Real-LOLBlur, we use non-reference metrics for evaluation.

For training with AIO-Blur, we equally scaled the size of the datasets to make sure that the network learned enough features for each dataset. The largest dataset is LOLBlur, with 10,200 pairs of images. As this is a very large compared with the other datasets, we worked on a reduction of this dataset to have a number of samples similar to the second largest dataset, RealBlur. The other datasets were upsampled to also have a similar number of samples like RealBlur. The final distribution of the different datasets in AIO-Blur can be seen in Table 4.

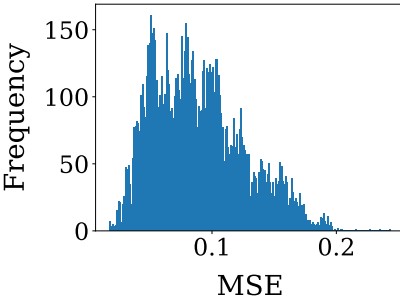 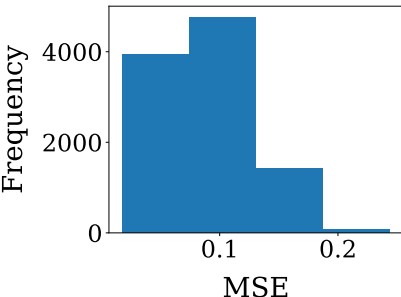

Figure 10: (Left) Distribution of MSE in LOLBlur. (Right) MSE-based classification for the downscaling of the dataset. Note that due its low population the last bin is not relevant in the whole distribution, thus the images that belong to this set are not considered in the final training set.

**Subsample of LOLBlur** To subsample this dataset, we calculate the mean-squared error (MSE) of all the images in the train split of the dataset. When checking the histogram of MSE values of all the images we found a large shift into small MSE values, which in most cases can be related to images that are easier to restore. To have a diverse set of images in this dataset, we draw the same histogram considering only four bins. Both of these histograms can be seen in Figure 10. Based on the low population of the last bin in the four-bin histogram, we did not considered images in this subset. Of the remaining 3 bins, we randomly picked 1,400 image pairs from each. The whole process led to the final 4200 pairs of training images of LOLBlur dataset in AIO-Blur dataset.

## C  IMPLEMENTATION DETAILS

Our implementation is based on PyTorch. We train DeMoE using the training set of AIO-Blur. We randomly cropped $384 \times 384$ patches and applied vertical and horizontal flip augmentations. The batch size is set to 32 and we used 4 H100 GPUs for training. The optimizer used is AdamW (Loshchilov & Hutter, 2017), setting $\beta_1 = 0.9$ and $\beta_2 = 0.9$, with an initial learning rate $1e^{-3}$ and updated to a minimum value of $1e^{-7}$ by the cosine annealing strategy (Loshchilov & Hutter, 2016). The training is divided into two steps: pretraining the baseline and finetuning the experts and decoder layers. Each of the steps has been trained for 400 epochs, for a total time of $\approx 6$ days.

## D  EXPERTS SPECIALIZATION

To show the specialization of each of the experts in the DeMoE network, in Figure 11 the correlations between the different experts are presented. Apart from the LayerNorm layers, the remaining layers of the five experts do not show significant correlation. Two ideas can be extracted from this analysis:

1. The normalization layers are very similar, so the features that are introduced in each of the experts are in a similar space.

2. Other layers do not share correlations because of the specialization of the different experts to each task. Thus, the MoEBlocks work as expected.

## E  CLASSIFICATION ERROR OF THE ROUTER

In Section 4.2, it has been pointed out how the router classification of the AIO-Blur-OOD is the one that produces the low results of DeMoE when manual expert selection is not used. To ilustrate this bad performance of the router, in Figure 12 we represent the average output tensor of the router for the different datasets on AIO-Blur and AIO-Blur-OOD. It can be seen that the expert usage for the AIO-Blur datasets is the expected one, while in the OOD case the router fails to classify the RSBlur dataset and Real-LOLBlur one.

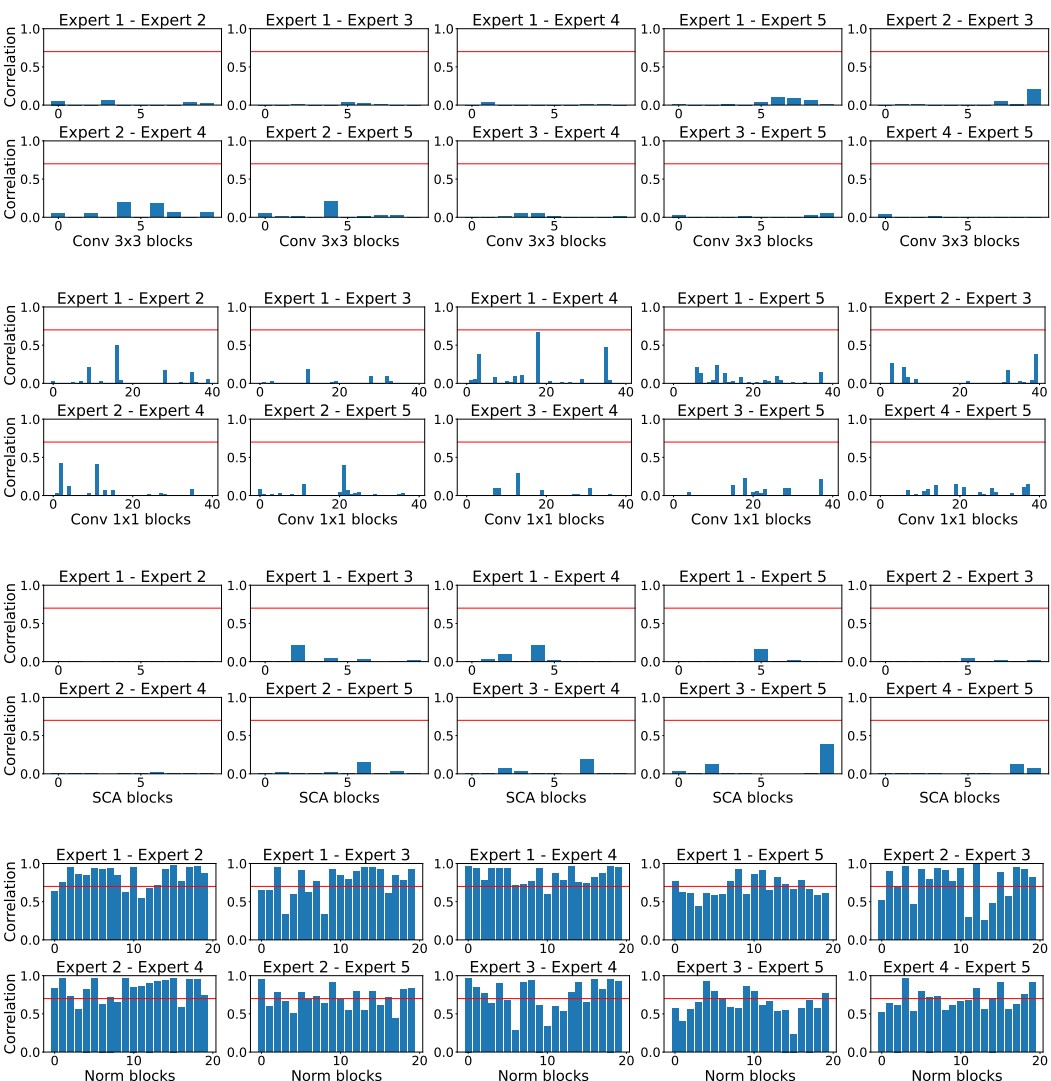

Figure 11: Correlations of the different experts in MoEBlock for the DeMoE network proposed. The lack of correlation in every layer, apart from normalization, suggest that the experts are specialized in the different deblurring tasks.

# F  FURTHER ABLATION STUDY

**Architecture Ablation**  In Table 5 (Left) we present the results of using different fusions of the generated features of each expert. The addition residual can be formulated as

$$\hat{\mathbf{h}} = \mathbf{h} + \sum_{i=0}^{N} \mathbf{w}_i \cdot \mathbf{e}_i(\mathbf{h}) \ , \tag{7}$$

where $\mathbf{h}$ and $\hat{\mathbf{h}}$ are the input and output features, respectively. $\mathbf{w}_i$ is the corresponding weight of the expert $\mathbf{e}_i$. Following the same formulation, the attention connection presented in Table 5 can be stated as

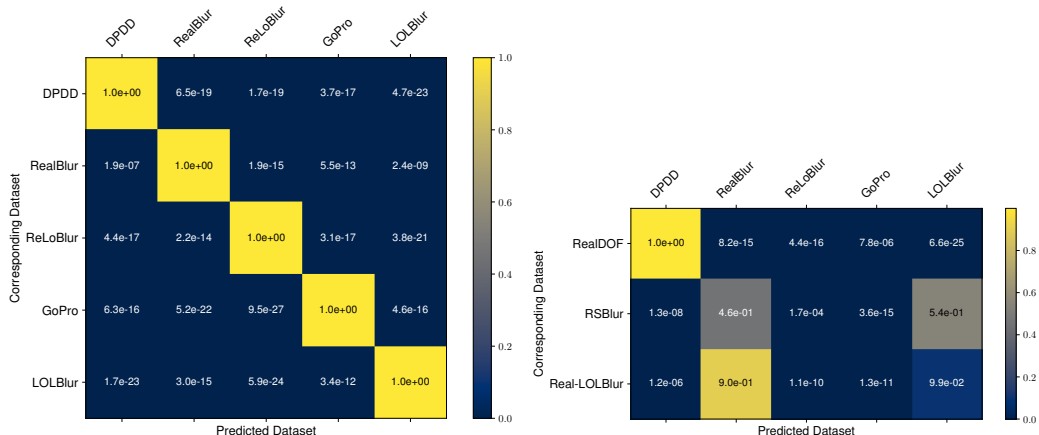

Figure 12: Average output tensor of the router for the AIO-Blur (Left) and AIO-Blur-OOD (Right) Datasets.

Table 5: (Left) Architectural ablations performed in DeMoE. (Right) Ablations performed in the pretrained baseline of DeMoE and number of blocks of DeMoE.

| Ablation | PSNR↑ | SSIM↑ | LPIPS↓ |
|---|---|---|---|
| Addition residual to MoEBlock output | 29.10 | 0.876 | 0.209 |
| Attention connection to MoEBlock output | 28.83 | 0.870 | 0.218 |
| Larger channel embedding ($32 \rightarrow 64$) | **29.23** | **0.880** | **0.203** |
| **DeMoE**$_{k=1}$ | 29.19 | 0.877 | 0.208 |

| Ablation | Params (M)↓ | MACs (G)↓ | Runtime (ms)↓ | PSNR↑ | SSIM↑ | LPIPS↓ |
|---|---|---|---|---|---|---|
| Baseline w/o classification loss | 10.08 | 11.02 | 8 | 28.84 | 0.868 | 0.225 |
| Baseline w/ original architecture | 22.07 | 20.37 | 14.8 | 29.19 | 0.878 | 0.207 |
| Baseline w/ classification loss | 10.08 | **11.02** | **8** | 29.12 | 0.873 | 0.215 |
| More middle NAFBlocks | 38.67 | 15.11 | 15.6 | 29.18 | 0.878 | 0.206 |
| More MoEBlocks | 32.54 | 13.18 | 17.4 | **29.26** | 0.879 | **0.203** |

$$\hat{\mathbf{h}} = \mathbf{h} \cdot \sum_{i=0}^{N} \mathbf{w}_i \cdot \mathrm{e}_i(\mathbf{h}) \quad . \tag{8}$$

We also trained a model with a larger embedding depth, increasing the channel count from 32 to 64. While this modification yielded slightly higher performance, the improvement was insufficient to justify the substantial increase in computational cost (79.71 M parameters, 42.77 G MACs, 22.4 ms runtime). Consequently, this model was not considered for further qualitative or out-of-distribution (OOD) analysis. Furthermore, we experimented with adding more NAFBlocks at the deepest encoder level and doubling the number of MoEBlocks in each decoder level. As shown in Table 5 (Right), these architectural expansions did not yield a more favorable trade-off. Collectively, these ablation studies confirm that the proposed DeMoE architecture achieves the optimal balance between performance and efficiency.

**Pretrained baseline ablations** Table 5 (Right) also presents ablations concerning the pretraining of the NAFNet baseline. Our proposed architecture has two NAFBlocks per each encoder layer and three for the middle block and decoder layers. This is different from the original design of NAFNet (Chen et al., 2022) for image deblurring, which typically employs one NAFBlock per encoder-decoder step, except for the final encoder step that uses 28 blocks. We evaluated the pretrained NAFNet using this original architecture (with three NAFBlocks per decoder layer) and include the results in Table 5. However, due to the substantial increase in operations, parameters, and runtime, we selected our more efficient baseline architecture.

In addition, we studied the impact of the classification loss on the pretrained baseline. The quantitative results can also be seen in Table 5, where it is shown that the inclusion of the classification loss notably increases the performance of the network. In Figure 13, we present some qualitative results on the use of the original baseline and classification loss. Given all the images in the test sets of the AIO-Blur dataset, we apply CLIP (Radford et al., 2021) and t-SNE (Van der Maaten & Hinton, 2008) in cascade to the encoder features of different weights: random initialized weights, AIO-Blur

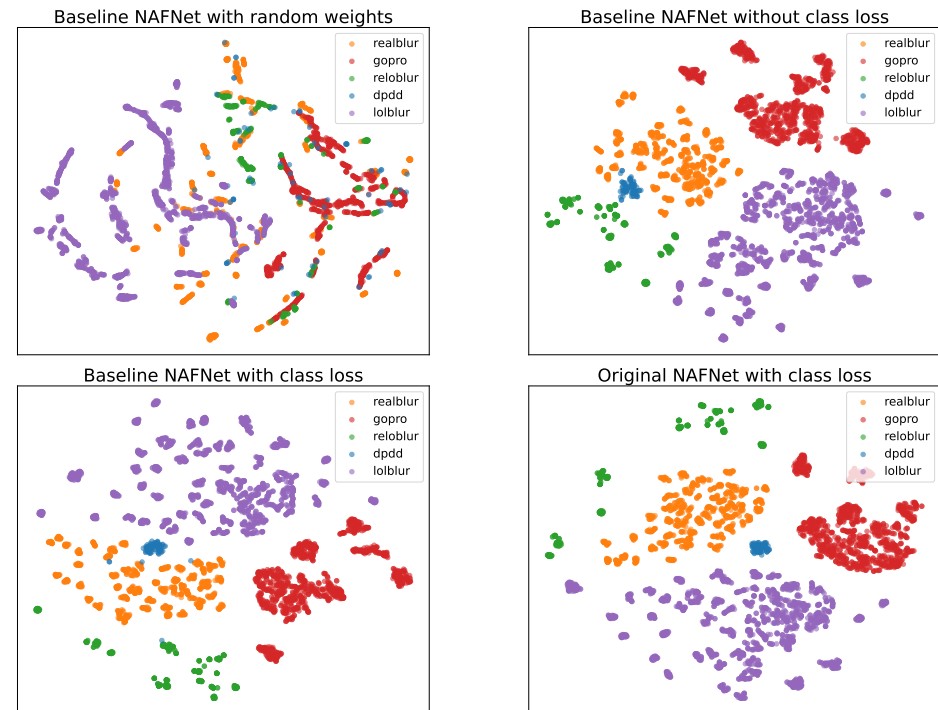

Figure 13: t-SNE representations of different baseline models.

weights without classification loss, AIO-Blur weights with classification loss, and original NAFNet architecture with classification loss. As expected, trained weights produce distinct clusters while random weights do not. Contrary to our expectation, the point clouds for all trained models appear remarkably similar. The key insight is that the similarity between the cluster plots of the original and our proposed architecture suggests that a heavily parameterized encoder stage is unnecessary. Instead, parameters are more effectively allocated to the task-specific restoration stage in the decoder. This finding reinforces the advantage of our proposed architecture over the original design.

**TLC Ablation** Table 6 presents the results of applying the test-time local converter (TLC (Chu et al., 2022)) to NAFNet blocks during inference. This method adapts the behavior of specific network layers at test time and is designed to improve performance on images larger than those used in training. While TLC leads to notable metric improvements in some restoration tasks, our study on its application to DeMoE considered three scenarios: (1) no TLC, (2) TLC applied to all layers, and (3) TLC applied partially only to experts where it yielded significant gains. The results in Table 6 indicate that TLC adversely affects the LOLBlur and ReLoBlur experts, leading to performance degradation in low-light and local-motion deblurring tasks. Since TLC did not provide a general improvement and even hampered performance on the low-light task, we excluded it from the final version of DeMoE.

**Experts Ablation** We conducted an ablation study on different expert blocks for constructing the MoEBlock. This exploration included modifications of the NAFBlock as well as other established architectures. Since deblurring requires a large receptive field, the selected blocks were based on this principle, utilizing dilated convolutions, large kernels, and transformer layers. Among these, the Restormer block Zamir et al. (2022b) achieved the highest performance but significantly increased the number of parameters and computational operations. We also evaluated the PLKBlock Lee et al. (2024), which employs large convolutional layers to expand the receptive field; however, as shown in Table 7, it performed poorly on the AIO-Blur dataset. To maintain the NAFBlock structure while increasing its receptive field, we incorporated the wavelet LWN block from Gao et al. (2024), which increased the model size by a factor of five. Finally, we tested the DarkIR block (Feijoo et al., 2025), originally designed for low-light enhancement. While its efficiency was comparable, its performance

Table 6: Comparison of the results of applying TLC during inference for NAFNet and DeMoE. It can be seen that some of the datasets metrics are improved when using TLC while others suffer a decrease. The average result suggests that it is better to not use TLC.

| Method | RealBlur | ReLoBlur | DPDD | LOLBlur | GoPro | Average |
|---|---|---|---|---|---|---|
| NAFNet | **29.18** / 0.883 / 0.203 | **34.56** / **0.926** / 0.228 | 25.60 / 0.795 / 0.266 | 26.69 / 0.871 / 0.188 | 29.56 / 0.890 / 0.191 | 29.12 / 0.873 / 0.215 |
| NAFNet w/ TLC | 29.15 / 0.884 / 0.199 | 34.33 / 0.925 / **0.227** | **25.77** / **0.806** / 0.258 | 25.80 / 0.873 / 0.185 | 29.95 / 0.897 / 0.184 | 29.00 / 0.877 / 0.211 |
| DeMoE$_{k=1}$ w/ TLC | 29.00 / **0.885** / **0.195** | 34.23 / 0.923 / 0.232 | 25.70 / **0.806** / 0.250 | 26.13 / **0.880** / 0.172 | **30.42** / **0.906** / **0.170** | 29.10 / 0.880 / 0.204 |
| DeMoE$_{k=1}$ w/ partial TLC | 29.00 / 0.885 / 0.195 | 34.26 / 0.924 / 0.232 | 25.70 / 0.806 / 0.250 | 26.32 / **0.881** / **0.172** | 30.42 / **0.906** / 0.170 | 29.14 / **0.880** / **0.204** |
| **DeMoE**$_{k=1}$ | 28.96 / 0.884 / 0.198 | 34.52 / 0.925 / 0.232 | 25.56 / 0.797 / 0.258 | **26.84** / 0.878 / 0.175 | 30.06 / 0.900 / **0.176** | **29.19** / 0.877 / 0.208 |

Table 7: A comparison of DeMoE's performance with different expert blocks on the AIO-Blur dataset shows that the considered NAFBlock architecture achieves the best trade-off between performance and efficiency. The metrics reported are average values across the dataset.

| Expert | Params (M)↓ | MACs (G)↓ | Runtime (ms)↓ | PSNR↑ | SSIM↑ | LPIPS↓ |
|---|---|---|---|---|---|---|
| DarkIR (Feijoo et al., 2025) | 21.9 | 11.05 | 14.2 | 29.15 | 0.876 | 0.207 |
| Wavelet Expert | 101.89 | 14.64 | 14.7 | 29.25 | 0.879 | 0.203 |
| Restormer (Zamir et al., 2022b) | 29.06 | 13.25 | 15.2 | **29.26** | **0.879** | **0.203** |
| PLK Expert (Lee et al., 2024) | 119.64 | 30.46 | 14.1 | 28.46 | 0.860 | 0.236 |
| DeMoE | **20.15** | **11.05** | **13.1** | 29.19 | 0.877 | 0.208 |

was inferior to the original NAFBlock. This ablation study confirms that the proposed architecture achieves the best performance/efficiency trade-off.

# G  MORE RESULTS IN OOD

**Quantitative results**   In addition to the quantitative results discussed in OOD in the main article, we present extended results in this dataset in Tables 8 and 9.

**Qualitative results**   We show the performance of DeMoe compared to the other general deblur methods using the OOD datasets in Figure 14. A state-of-the-art method is also included in each of the qualitative samples.

# H  BROADER IMPACTS OF DeMoE

As a preliminary exploration of task-related restoration, DeMoE has the following impacts:

- Applications in many fields: Compared to existing methods, DeMoE offers higher robustness to different scenarios where blurry artifacts can be generated. It can be widely applied to any computer vision task with images that can potentially suffer blur degradation, such as autonomous driving or commercial photography.

- Negative social impacts: To the best of the authors knowledge, there are no negative social impacts.

Table 8: Quantitative evaluations on various datasets. Results with † were extracted from (Lee et al., 2021)(RealDOF) and (Gao et al., 2024)(RSBlur). Results with * were trained in the AIO-Blur dataset.

| Method | PSNR↑ | SSIM↑ | LPIPS↓ |
|---|---|---|---|
| Restormer* (Zamir et al., 2022a) | 22.98 | 0.684 | 0.471 |
| FFTFormer* (Kong et al., 2023) | 23.36 | 0.697 | 0.440 |
| SFHFormer* (Jiang et al., 2024) | 24.20 | 0.736 | 0.428 |
| NAFNet* (Chen et al., 2022) | 24.64 | **0.762** | 0.382 |
| JNB† (Shi et al., 2015) | 22.36 | 0.635 | 0.601 |
| EBDB† (Karaali & Jung, 2017) | 22.38 | 0.638 | 0.594 |
| DMENet† (Lee et al., 2019) | 22.41 | 0.639 | 0.597 |
| DPDNet† (Abuolaim & Brown, 2020) | 22.67 | 0.666 | 0.420 |
| IFAN† (Lee et al., 2021) | **24.71** | 0.748 | **0.306** |
| **DeMoE**$_{k=1}$ | 24.59 | 0.758 | 0.377 |

**RealDOF** results

| Method | PSNR↑ | SSIM↑ |
|---|---|---|
| Restormer* (Zamir et al., 2022a) | 14.04 | 0.556 |
| SFHFormer* (Jiang et al., 2024) | 13.54 | 0.553 |
| NAFNet* (Chen et al., 2022) | 15.00 | 0.576 |
| FFTFormer* (Kong et al., 2023) | 15.60 | 0.589 |
| FFTFormer† (Kong et al., 2023) | 29.70 | 0.787 |
| BANet+† (Tsai et al., 2022b) | 30.24 | 0.809 |
| MLWNet-B† (Gao et al., 2024) | **30.91** | **0.818** |
| **DeMoE**$_{k=1}$ | 19.23 | 0.640 |
| **DeMoE**$_{k=realblur}$ | 27.53 | 0.763 |

**RSBlur** results

Table 9: Robustness study of OOD night blurry images using Real-LOLBlur dataset (Zhou et al., 2022). Methods with * were trained in AIO-Blur dataset, but methods with † were extracted from Zhou et al. (2022).

| | RUAS → MIMO† | MIMO → Zero-DCE† | FFTFormer* | NAFNet* | Restormer* | SFHFormer* | LEDNet† | DarkIR | DeMoE$_{k=1}$ | DeMoE$_{k=lolblur}$ |
|---|---|---|---|---|---|---|---|---|---|---|
| MUSIQ↑ | 34.39 | 28.36 | 30.18 | 32.02 | 27.71 | 29.58 | 39.11 | **48.36** | 31.34 | 38.91 |
| NRQM↑ | 3.322 | 3.697 | 5.780 | 5.515 | 5.169 | 5.320 | 5.643 | 4.983 | 5.47 | **6.113** |
| NIQE↓ | 6.812 | 6.892 | 6.469 | 6.865 | 7.381 | 6.806 | **4.764** | 4.998 | 7.56 | 5.82 |

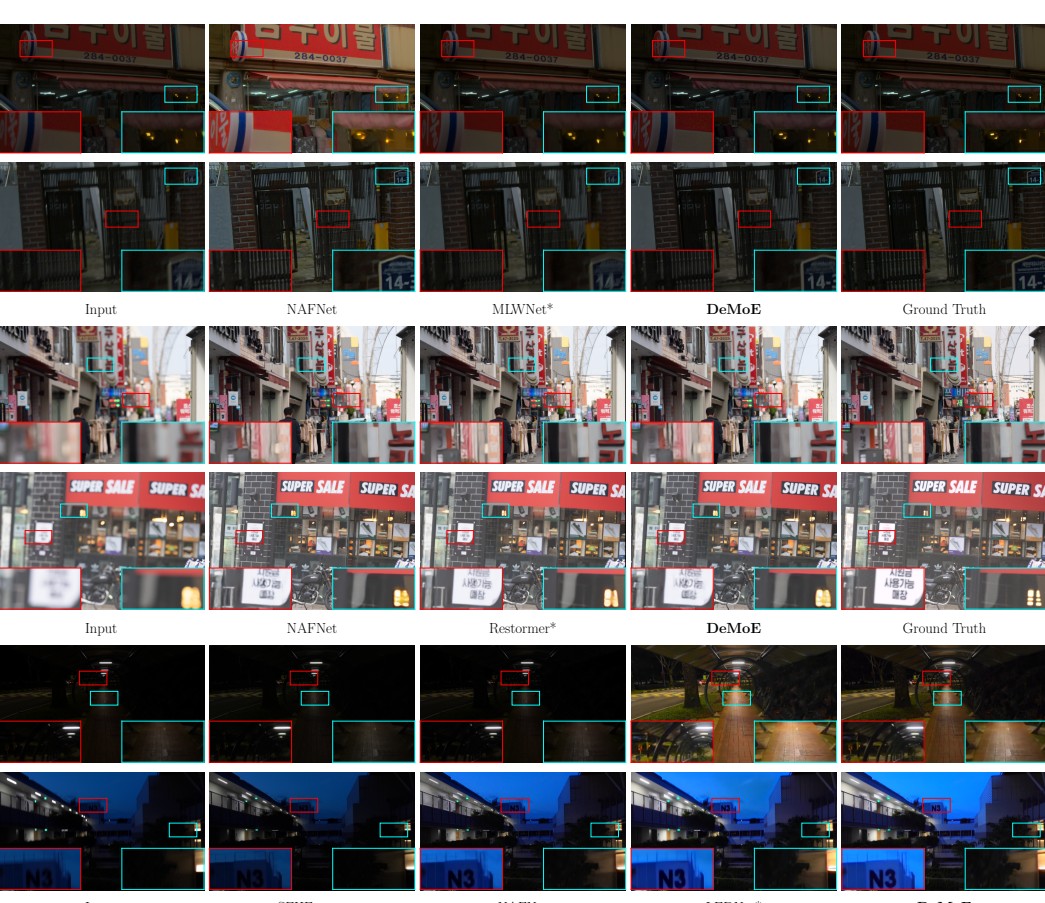

Figure 14: Qualitative comparison of the general deblur methods in OOD datasets. Methods with * are task-specific ones. The first two rows are images from RSBlur (Rim et al., 2022), the next two rows are from RealDOF (Lee et al., 2021), and the final rows are from Real-LOLBlur (Zhou et al., 2022). Zoom in for better view.

# I LLM DISCLOSURE

During the preparation of this manuscript, the authors utilized large language models (LLMs) exclusively for proofreading and grammatical refinement. All scientific content, analysis, and intellectual contributions remain entirely human-authored.

