# OpenReview forum: "Towards Unified Image Deblurring using a Mixture-of-Experts Decoder"
_ICLR.cc/2026/Conference — ICLR 2026 Conference Withdrawn Submission_

### Official Review · Reviewer_SeTP · 2025-10-25

**Soundness:** 3
**Presentation:** 3
**Contribution:** 2
**Rating:** 4
**Confidence:** 5

**Summary:**

This paper proposes the first all-in-one deblurring method capable of efficiently restoring images affected by diverse blur degradations, including global motion, local motion, blur in low-light conditions, and defocus blur. To achieve this, the authors propose a mixture-of-experts (MoE) decoding module, which dynamically routes image features based on the recognized blur degradation, enabling precise and efficient
restoration in an end-to-end manner. The proposed approach not only achieves performance comparable to dedicated task-specific models, but also shows promising generalization to unseen blur scenarios.

**Strengths:**

1. This paper proposes the first all-in-one deblurring method that can efficiently restore any blurry image. The motivation and contributions are technically sound, and the design of the mixture-of-experts (MoE) decoding module is novel.

2. The subsection “Deblurring Similarity Analysis” effectively introduces the importance of an all-in-one network for deblurring, enabling readers to easily understand the motivation behind the proposed method.

3. The paper is well written and easy to follow.

**Weaknesses:**

1. Although the motivation and idea are technically sound, the performance of the proposed DeMoE seems unsatisfactory. As shown in Table 1, the performance of DeMoE$_{k=1}$ works similar to the baseline NAFNet while with two times number of parameters.

2. As the authors propose a new all-in-one model, it would be better to compare the proposed DeMoE with more recent all-in-one methods in Table 1, such as AdaIR [A] (ICLR 2025), MoCE-IR [B] (CVPR 2025), and DFPIR [C] (CVPR 2025).

3. Although the authors claim to use multiple experts, the experimental results in Tables 1, 2, and 3 only report the performance of a single expert, DeMoE$_{k=1}$, which appears somewhat inconsistent with the proposed multi-expert framework.

[A] AdaIR: Adaptive All-in-One Image Restoration via Frequency Mining and Modulation. In CVPR 2025.

[B] Complexity Experts are Task-Discriminative Learners for Any Image Restoration: . In CVPR 2025.

[C] Degradation-Aware Feature Perturbation for All-in-One Image Restoration. In CVPR 2025

**Questions:**

1. In Tables 1, 2, and 3, why do you report only the performance of a single expert, DeMoE$_{k=1}$.

 How about experts such as DeMoE$_{k=5}$?

2. Could your compare the proposed method with the latest all-in-one models, such as AdaIR [A] (ICLR 2025), MoCE-IR [B] (CVPR 2025), and DFPIR [C] (CVPR 2025).

3. I am curious about how the proposed method handles images that contain both motion blur and defocus blur. In some cases, one may wish to preserve the defocus blur while removing only the motion blur. However, your method appears to remove both types of blur simultaneously.

[A] AdaIR: Adaptive All-in-One Image Restoration via Frequency Mining and Modulation. In CVPR 2025.

[B] Complexity Experts are Task-Discriminative Learners for Any Image Restoration: . In CVPR 2025.

[C] Degradation-Aware Feature Perturbation for All-in-One Image Restoration. In CVPR 2025

---

> ### Author Response · Authors · 2025-11-14
> **Author Response to Reviewer SeTP**
>
> We sincerely thank the reviewer for their careful and insightful assessment of our work. We greatly appreciate the reviewer highlighting the novelty of DeMoE as the first all-in-one deblurring method, the technical soundness of our contributions, and the clear presentation of the “Deblurring Similarity Analysis” subsection. The reviewer’s comments have helped us clarify key aspects of our method and its experimental validation.
> Below, we address all weaknesses and questions raised, providing clarifications and additional context.
>
> 1. DeMoE vs NAFNet performance
> - **The performance of DeMoE surpasses NAFNet** not only on the training datasets but also on **OOD datasets**. With manual expert selection, DeMoE achieves good restoration, while   NAFNet struggles on RSBlur and Real-LOLBlur.
> - While DeMoE has twice the parameters of NAFNet, **not all are used during inference with DeMoE_k=1**. This is reflected in similar runtime and number of operations, indicating that the extra parameters are primarily for training.
>
> 2. Additional comparisons
> - Thanks for the suggestion. We will consider training these SOTA all-in-one models for comparison. Note that these methods are not specifically designed for deblurring, so their performance is expected to be lower than DeMoE and other general deblurring models we trained (Restormer, FFTFormer, SFHFormer, NAFNet). **We focus on the most popular deblurring models i.e., optimal backbones for this task.**
>
> 3. Number of used experts
> - This is because no datasets contain multiple blur degradations in the same image. The best performance is obtained using the expert corresponding to the specific degradation. To demonstrate the benefits of multiple experts on OOD data, a dataset with combined degradations would be required. We plan to work on this in future.
>
> **Questions & Responses**
>
> 1- Performance reports of DeMoE k = 5: This has been addressed in Weakness 3.
>
> 2. Additional Comparisons: This has been addressed in Weakness 2.
>
> 3. Handling diverse types of blur in images: Currently, the method removes all blur present in the image. It cannot selectively remove motion blur while preserving others. This is an interesting direction for future work.
> We believe that there are no datasets to verify this (maybe Bokeh defocus with the subject moving) -- we would appreciate suggestions.
>
> We hope that these clarifications address the reviewer’s concerns and provide a clearer understanding of DeMoE’s design, capabilities, and evaluation.

---

### Official Review · Reviewer_1g1q · 2025-10-31

**Soundness:** 2
**Presentation:** 3
**Contribution:** 2
**Rating:** 2
**Confidence:** 4

**Summary:**

This paper aims to propose a unified image deblurring model to tackle various blur types in the real world. The paper first investigated the blur type difference of deblurring datasets by the network similarity study.  Then, a mixture-of-experts decoder is introduced to deal with various blur types in a divide-and-conquer way. The performance is evaluated across several deblurring benchmarks, including synthetic and real blur scenarios.

**Strengths:**

- Network similarity study provides a novel and insightful analysis of the blur types in different deblurring datasets. The conclusion also contributes to the community.
- The MoE structure yields an efficient method compared with previous sotas.
- The paper is well-written and easy to follow.

**Weaknesses:**

- The effect of general deblurring is not satisfactory. In Table 2, DeMoE without manual selection cannot surpass previous methods in both RealDOF and Real-LOLBlur. Meanwhile, the reviewer considers manually selecting experts as a task-specific method, since the type of blur in the input image should be unknown in the unified deblurring scenario.
- The MoE router is trained with the ground-truth degradation label. Where does the label come from?
- The allocation of the router in the MoE module needs to be analyzed using various deblurring datasets to verify whether it draws consistent conclusions with the network similarity study section.
- DeMoE is trained by the proposed AIO-Blur dataset. How about other rivals listed in the experimental results? A fair comparison is supposed to use the same training data.

**Questions:**

Please refer to the weakness part.

---

> ### Author Response · Authors · 2025-11-14
> **Author Response to Reviewer 1g1q**
>
> We appreciate the reviewer's detailed feedback, which helps us improve the clarity and rigor of our work. We are pleased that the reviewer recognizes the Strengths of our method, including the comprehensive experiments and the practical significance of DeMoE as a unified framework. We have revised our manuscript to address all Weaknesses and Questions raised. We will address the reviewers questions and answers in the same order that were proposed.
>
> 1) About Unified model and Task-specific concern
> We respectfully disagree with the comment/claim. DeMoE is the first unified (all-in-one) deblurring method. We tested our single model on 8 benchmarks, in some, **the general model matches task-specific methods** -- see LOLBlur, ReLoBlur, DPDD.
>
> - The manual selection experiment was included to quantify the inherent capability of each expert to handle specific degradation types, similar to ablation studies in other all-in-one restoration literature.
> - Our router learns to infer the degradation type and allocate the most appropriate expert(s), without any prior information i.e, blind degradation prediction and restoration, based only on the input.
>
> We have added a section to the discussion in the revised manuscript to better position DeMoE within the literature on degradation-aware all-in-one image restoration. \
>
> 2) About Router Training \
> The reviewer asks for the source of the ground-truth degradation label used for training the MoE router. The degradation label is derived from the original dataset category that the image belongs to. We clarified this in the paper.
>
> 3) About Router Allocation Analysis \
> The reviewer requests an analysis of the router's allocation consistency across different datasets, to verify the findings of the network similarity study.
> **We believe this analysis is already provided in the supplementary material.**
> The expert similarity analysis presented in Figure 7 of Section A (Supplementary Material) provides a quantitative measure of the weight differences between experts. This analysis directly supports the motivation for the MoE design by showing that experts do not converge to the same parameters, indicating that they have learned to specialize.
>
> 4) About Training Data Fairness \
> As we mention in the paper **Lines 317-318; " For fairness, all methods are trained on the AIO-Blur dataset".** -- Section 4.1 (Training Details).
> We confirm that all competing methods reported in Table 1 and the General Methods reported in Table 2 were trained from scratch using the exact same AIO-Blur training dataset that we proposed and used to train DeMoE.
> We have highlighted this statement in bold in the revised manuscript to prevent any future misunderstanding.

---

### Official Review · Reviewer_uDGq · 2025-11-01

**Soundness:** 2
**Presentation:** 3
**Contribution:** 2
**Rating:** 2
**Confidence:** 5

**Summary:**

This paper proposes a unified image deblurring framework, DeMoE (Mixture-of-Experts Decoder), which employs a mixture-of-experts mechanism to handle multiple blur types within a single model. Through network similarity analysis, the authors find strong parameter correlations among task-specific deblurring models, motivating a router-controlled MoE decoder that dynamically selects experts for different blur types. Additionally, the paper constructs the AIO-Blur dataset, integrating multiple blur scenarios for unified training and evaluation.

**Strengths:**

1. Comprehensive experiments: The study includes AIO-Blur and OOD testing, task-specific comparisons, ablation studies, and efficiency analyses, offering broad coverage and credible conclusions.
2. Practical significance: DeMoE serves as a unified framework applicable to diverse blur scenarios, reducing the need for multiple specialized models and showing potential for real-world deployment.

**Weaknesses:**

1. This article lacks significant innovation, and its multi expert strategy is very common in all-in-one image restoration, only changing the focus of different restoration tasks in all-in-one image restoration to different scenes under the single task of deblurring.

2. Lack of algorithmic overview: Although Figure 3 shows the network architecture, a concise workflow summarizing differences between training and inference stages is missing.

3. Limited router generalization: The router performs poorly on OOD datasets, leading to incorrect expert selection. While manual expert control mitigates this, it undermines the “dynamically unified model” objective.

4. Unclear expert sharing and independence mechanism : The paper does not clarify whether expert parameters are fully independent or partially shared, nor analyze how this design affects model capacity and generalization.

5. In Table 1, DeMoE fails to reach SOTA on nearly half of the six datasets. On RealBlur, DeMoE is inferior to SFHFormer which has less parameters; on ReLoBlur and DPDD, DeMoE shows no advantage over NAFNet which has less Computational Cost.

6. In Table 3, DeMoE performs near the bottom among compared methods, indicating unsatisfactory results.

7. Labeling error: In Table 3 (ReLoBlur results), two different SSIM values are both marked as SOTA. SSIM(LBAG)=0.9249 and SSIM(DeMoEk=1)=0.925, but the former is marked as 2nd best.

**Questions:**

1. What are the differences between DeMoE’s training and inference stages? Consider adding pseudocode or a system flowchart for clarity.

2. Has the router’s classification accuracy been evaluated on OOD datasets? Would uncertainty-based or entropy-based gating improve its generalization?

3. Since the encoder is shared while each expert in the decoder has independent convolutional modules, please analyze this “partially shared + partially independent” design trade-off. Specifically, does full independence improve performance? Could partial sharing enhance generalization or parameter efficiency? An ablation or comparative study is recommended.

---

> ### Author Response · Authors · 2025-11-14
> **Author Response to Reviewer uDGq**
>
> We sincerely thank the reviewer for the thorough and critical assessment of our work. We greatly appreciate the reviewer’s deep engagement with the field and the high confidence shown in evaluating our manuscript. The review has highlighted several areas where we can provide additional clarification and improvements. We are pleased that the reviewer recognizes the comprehensive experiments and practical significance of DeMoE.
>
> Below, we address all weaknesses and questions raised, incorporating clarifications and additional analyses in the revised manuscript and supplementary material.
>
> 1. Lack of significant innovation.
>        Our method is the first all-in-one deblurring model addressing multiple blur types simultaneously. As shown in Figure 1, models trained for a specific blur (e.g., defocus) fail on other types. This motivates the need for a unified model, which our DeMoE provides.
> 2. Lack of algorithmic overview.
>        Figure 3 illustrates the experts’ usage during both training and inference, as further detailed in Section 3.3. We will emphasize this in the final version.
> 3. Limited router generalization.
>        We acknowledge this limitation, caused by limited diversity in the training datasets. Future work will expand the dataset to improve generalization. Despite this, our contributions—proposing all-in-one deblurring, gathering datasets, designing a training pipeline, and creating a suitable architecture—remain significant.
>  4. Unclear expert sharing and independence mechanism.
>        Experts are fully independent. While this was implied, we will clarify it in the final version. Section D of the supplementary material provides an analysis demonstrating their independence.
>  5. Concerns about Table 1.
>       There are 5 datasets: DeMoE achieves SOTA on 3 and second-best on 2. On average, DeMoE outperforms others. **While SFHFormer has fewer parameters, DeMoE is 3× faster, requires 5× fewer operations**, and only uses 10.08M parameters per expert during inference. Compared to NAFNet, DeMoE maintains efficiency and superior OOD generalization.
>  6. Concerns about Table 3.
>         DeMoE ranks among the top three methods on 3 out of 5 datasets (LOLBlur, ReLoBlur, DPDD), contradicting your statement. Moreover, **It is well-known that All-in-one models generally underperform task-specific models.**, specially when we consider several tasks and real-world evaluation.
>  7. Labeling error in Table 3.
>         We corrected the typo in Table 3.
>
> Questions
> 1. Differences between training and inference stages.
> Figure 3 shows the workflow. Training uses all experts; inference selects only the top-K experts (Section 3.3).
>
> 2. Router evaluation on OOD datasets.
> Yes, see Section E and Figure 12 in the supplementary material. Gaussian noise during training was tested but ineffective. The main limitation is dataset diversity; future work will address this.
>
> 3. Analysis of “partially shared + partially independent” design.
> Full independence improves performance, as shown in DeMoE*. Using one expert (similar to a single-task NAFNet) gives slightly higher PSNR (0.3% on average), but DeMoE reduces parameters from 50.4M to 20.15M (2.5× fewer) with minimal performance loss, achieving a better efficiency-performance trade-off.
>
> We hope these clarifications address the reviewer’s concerns and strengthen the manuscript

---

### Note · Authors · 2025-11-14

**Comment:**

We appreciate the time and effort from the reviewers, AC and PC members, specially Reviewer SeTP.

However, most pointed weaknesses were addressed and even explained in the main paper.
The main weakness is actually highlighted by us in the paper, Line 409: our all-in-one method **--the first all-in-one method for real-world deblurring--** struggles to surpass (in every benchmark) task specific models designed and trained on particular dataset.

> All-in-One (AIO) restoration methods typically **underperform** compared to task-specific methods. For instance, AIO Restormer achieves a PSNR (dB) of  27.76 in GoPro deblurring, while the task-specific version achieves 32.92 (*+5.2dB*). This is a **well-known practical limitation of AIO methods** Thus, comparison with task-specific methods is only for reference purposes (i.e., upper bound of the solution).

Surprisingly, **our all-in-one restoration method was matching task-specific SOTA on three datasets** (LOLBlur, ReLoBlur, DPDD), *We hope that someone could point out to a single publication where an AIO method outperforms all task-specific methods in 5 tasks.*


We appreciate that the reviewers found interesting our Deblurring Similarity Analysis, and acknowledge the efficiency and practical benefits of DeMoE --- equivalent to 5 task-specific NAFNets of 10M parameters.

Hope you understand our point.

**Withdrawal Confirmation:**

I have read and agree with the venue's withdrawal policy on behalf of myself and my co-authors.